# Diagnosis from Tissue: Histology and Identification

**DOI:** 10.3390/jof8050505

**Published:** 2022-05-13

**Authors:** Raquel Sabino, Nathan Wiederhold

**Affiliations:** 1Reference Unit for Parasitic and Fungal Infections, Department of Infectious Diseases, National Institute of Health Dr. Ricardo Jorge, 1649-016 Lisbon, Portugal; 2Instituto de Saúde Ambiental, Faculdade de Medicina, Universidade de Lisboa, 1649-028 Lisboa, Portugal; 3Fungus Testing Laboratory, Department of Pathology and Laboratory Medicine, University of Texas Health Science Center at San Antonio, 7703 Floyd Curl Drive, San Antonio, TX 78229, USA; wiederholdn@uthscsa.edu

**Keywords:** histopathology, fungal infection, tissue sample, laboratorial diagnosis

## Abstract

The diagnosis and initiation of appropriate treatment against invasive fungal infections depend upon accurate identification of pathogens by pathologists and clinical microbiologists. Histopathology is often critical in providing diagnostic insight in patients with suspected fungal infections, and such findings are incorporated into the definitions of proven or probable disease caused by certain pathogens. Such examinations can offer provisional identifications of fungal organisms, which can help guide initial therapy while laboratory results are pending. Common etiologic agents of invasive mycoses may be recognized based on morphologic characteristics observed in tissue and biologic fluids, such as those obtained from bronchoalveolar lavage and bronchial washings. However, care should be taken in the interpretation of these findings, as there may be a false sense of the ability to correctly categorize fungal organisms to the genus or species level by morphologic features alone. Studies have demonstrated discordant results between histopathology and laboratory results due to overlapping morphologic features, morphologic mimics, and sampling errors. Thus, histopathology plays an integral role in providing a differential of potential fungal pathogens but must be combined with results from laboratory studies, including cultures, antigen tests, serology, and molecular assays, in order to improve accuracy in the identification of etiologic agents of fungal infections. Inaccurate identification of the infecting organism can lead to inappropriate antifungal therapy and possibly poor clinical outcomes.

## 1. Introduction

Many fungal species are ubiquitous in nature and are widely distributed in soil, plant debris, and other organic substrates. Although it has been estimated that more than 12 million different of fungal species may exist [1,2,3], a very small percentage have been fully described. Approximately 700 are currently known to cause infections in humans [4,5]. The incidence of invasive fungal infections is rising due to advances in medical care and increases in at-risk patient groups [6]. While significant improvements have been made in disease management and the treatment of fungal infections, fungi substantially contribute to patient mortality. Recent studies have estimated that fungal infections may be responsible for 1.5 million deaths per year globally [7]. Early diagnosis of many invasive fungal infections is challenging but key for improving patient outcomes [8]. Rapid and accurate diagnosis of invasive fungal disease is associated with improved patient outcomes, including reductions in mortality as well as the reduced use of unnecessary antifungal therapy. Discontinuation of inappropriate or unnecessary therapy can help to limit drug–drug interactions and adverse effects and toxicities that are often associated with clinically available antifungals, especially in certain groups at risk for fungal infections that often receive polypharmacy. Current methods for diagnosing invasive fungal infections include assessment of clinical signs and symptoms, imaging procedures, cytological and/or histological examination of biopsy material, and laboratory tests, including fungal cultures and non-culture-based techniques (e.g., antigen and antibody detection, and molecular methods) [9]. Here we briefly discuss the role of histopathology in the diagnosis of invasive fungal diseases and its limitations in the identification of the etiologic agent of these infections.

## 2. Use of Fungal Cultures for Diagnosis

For the diagnosis of invasive fungal infections, culture has been considered the gold standard for laboratory diagnosis. An invasive fungal infection is considered proven if a positive culture is obtained from a sterile sample. A positive culture also suggests a probable infection when there are multiple isolations from non-sterile sites. Cultures also allow for identification of the etiological agent to the section or species level (currently performed by assessment of phenotypic characteristics, as well as molecular (e.g., PCR assays and DNA sequence analysis) and proteomic-based methods (e.g., MALDI-TOF MS)), for phenotypic susceptibility testing, and genotyping for epidemiologic purposes. However, fungal cultures have several limitations. The sensitivity of yeast cultures from the bloodstream has been reported to range between 21–71% due to the rapid clearance of viable organisms, leading to a relatively low number of colony-forming units per milliliter of blood [10,11]. In addition, these cultures often take 2 to 3 days before results become available, and thus the information is not available to clinicians in real-time. Mold cultures from respiratory tract specimens are even less sensitive even in confirmed cases of invasive disease [12,13,14,15,16]. In a study conducted at MD Anderson Cancer Center, cultures were positive in only 53% of autopsy-proven cases of invasive fungal infections. This rate was similar to the 58% culture positivity rate with cytologic evidence of fungi, and higher than the 30% culture positivity rate with histologic evidence of fungal disease [16]. Furthermore, cultures from the upper respiratory tracts often reflect colonization rather than invasive disease. Due to the ubiquitous nature of certain filamentous fungi, cultures are also subject to spurious contamination by conidia/spores in the environment [17,18]. Mold cultures also fail to yield information in real-time. Several factors may influence the sensitivity of fungal cultures, including volume or amount of specimen submitted to the laboratory and the concentration of fungi within the specimen, the method of specimen processing by laboratory personnel (e.g., homogenization vs. mincing, which is important for successful culture of Mucorales), the growth medium used, and the length and temperature of incubation. In addition, the sensitivity of cultures as well as other diagnostic assays may be reduced in patients receiving antifungal therapy [19,20,21].

Over the last decade there have been marked advances in the non-culture-based diagnostic assays. Several characteristics are sought in such assays, including the ability to detect pathogens early during the course of infection, good clinical sensitivity and specificity, broad discrimination among species, and the ability to quantitate fungal burden, which may help distinguish between colonization and disease [22]. Currently, no one diagnostic platform meets all of these criteria. Many molecular based-platforms that are that are performed on direct specimens are not widely available in clinical microbiology laboratories, nor is any one of these currently sufficient to replace cultures as the gold-standard for the diagnosis of invasive mycoses [18].

## 3. Use of Histopathology for Diagnosis of Invasive Fungal Infections

Histopathology continues to be a rapid and cost-effective means of providing a presumptive diagnosis of invasive fungal infections, as the presence of fungi, as determined by histopathology or cytology within normally sterile tissue or fluids, is also a criterion for proven fungal infection in patients with risk factors for invasive infections [9]. In many instances, histopathology may be the only means of diagnosis when material is not submitted for culture or other laboratory tests, or when the results of such laboratory evaluations are still pending [18]. Histopathology also has the advantages of being standardized and relatively cost-effective, although invasive procedures are often required to obtain the needed tissue. Several factors that should be considered during histopathology work-up and which affect the differential diagnosis include the host immune status, the patient’s travel and social history, the geographic region when infection may have occurred, and antifungal exposure [18,23]. To identify and assess the morphology of fungus in tissue sections, special stains are better than the routine hematoxylin eosin (H&E) stain. Two special stains for fungi include Gomori’s methenamine silver (GMS) and periodic acid–Schiff (PAS) stains [17,24]. GMS with H&E counterstain can demonstrate both fungi and tissue reactions, which may help the histopathologist. However, GMS and PAS alone cannot show the associated reactive tissue response [25].

Histopathologic evaluations are often combined with laboratory-based diagnostics, including antigen tests, serology, and molecular-based assays, in order to provide information to clinicians, including the presence or absence of tissue invasion and the degree of inflammatory response elicited by fungi, which can help to distinguish contamination or colonization from true infection. The results of molecular-based tests are most valuable in histopathologically proven infection but may have marginal diagnostic yields when fungal elements are not present [26,27,28,29,30,31]. Because of this, the most recent EORTC/MSG criteria for the classification of invasive fungal disease recommend that fungal PCR assays, with or without DNA sequencing, be performed on tissue when fungal elements are seen by histopathology [9].

To help establish a diagnosis of invasive fungal disease, histopathology findings should take into account the fungal morphologies that are observed combined with the extent of tissue invasion and the immune response seen within the specimen [17,18,23]. Morphologically, fungi may appear as yeast forms, hyphal forms (molds), or as dimorphic forms. Morphological parameters that may help to differentiate between different types of fungi on histology are the fungal structures’ size, the predominant form of the fungus (yeast, hyphae or pseudohyphae), presence or absence and thickness of the capsule, single or multiple budding (with narrow or wide base), the branching pattern observed with hyphae, and the presence or absence of septae. In yeast forms, fungi may be seen either in short chains or are individually separated. Yeast cells may be round to oval and budding may be observed. In certain *Candida* species, the buds often do not detach from each other and a chain of elongated yeast cells (pseudohyphae) can be observed [17,23]. The thick polysaccharide capsule of *Cryptococcus* gives to these organisms the characteristic appearance of having a clear space surrounding them that can be seen in tissue sections with H&E stains. As with all other yeasts, the wall of the organism stains with GMS and PAS stains. In addition, cryptococci stain with Fontana–Masson stain because they contain melanin. The polysaccharide capsule, specifically, stains with Alcian blue and Mayer’s or Southgate’s mucicarmine stain [17].

For invasive mold infections, hyphae are usually only observed within tissue. However, vesicles and conidia may be observed in cavitary lesions within the lung or sinuses [17,18,23]. Hyphae within tissues are generally classified into three broad categories, including: (1) hyaline septate, such as *Aspergillus*, *Fusarium*, and *Scedosporium* among others; (2) hyaline pauci-septate, such as members of the orders Mucorales and Entomophthorales; and (3) pigmented or dematiaceous, which can include members of the genera *Madurella*, *Fonsecaea*, *Cladophialophora*, *Bipolaris/Curvularia*, *Exophiala*, *Alternaria*, and others [17,23]. In tissues infected with dematiaceous fungi, Fontana–Masson staining may be useful to detect melanin [17]. While hyphae in tissues signifies proven invasive fungal infections in patients at risk, the morphologies observed are not able to provide a definitive identification to the genus or species level [18,32,33]. For example, despite distinct differences in appearance in tissue between the Mucorales (broad, aseptate or pauciseptate hyphae with non-dichotomous branching) and hyaline septate molds, such as *Aspergillus*, overlap of morphologies can occur when hyphae are scant, folded, or fragmented, as well as in areas of necrosis [17,23,32]. Misidentification can also occur with different yeasts. For example, there can be significant overlap in appearances between *Histoplasma* when found extracellularly and other smaller yeasts, including different *Candida* species (e.g., *Candida glabrata*) [17,23,32]. Fungal morphology may be distorted due to the type of biopsy, the quality of the specimen, and the host’s immune status. Though various stains are useful in delineating fungal morphology, they have limitations when the material is scanty, and the number of fungal elements is sparse. These limitations may lead to misinterpretation and misdiagnosis based on histopathology [34]. Therefore, highly experienced histopathologists are essential to detect fungal structures and also to recognize tissue reactions associated with invasive fungal disease, distinguishing them from staining artifacts and tissue reactions.

In addition to being knowledgeable about fungal morphology within tissues, there must be an understanding of tissue reactions in response to different infections. Different tissue reactions, including eosinophilic infiltrations, xanthogranulomatous reactions, and foreign body granulomas with or without necrosis, provide clues for the histopathologist to look for fungi in tissue sections [35]. The tissue reaction depends on the type of fungus, site involved, and immune status of the host.

## 4. Accuracy of Histopathology for Identification of Etiologic Agent

Microscopic observation of distinctive structures compatible with dimorphic fungi, such as *Blastomyces*, *Paracoccidioides*, and *Coccidioides* spp., is a criterion for proven fungal infection, with immediate identification of the etiological agent [9]. *Blastomyces* in tissue appears as multinucleate yeasts that measure 8 to 15 cm in diameter, have thick refractile cell walls, and may show a single, broad-based bud. Exceptions include *B. helicus*, which tends to form multiple buds that can appear as branching chains, and *B. percursus*, where short hyphal-like fragments can accompany yeast cells. In tissues infected with *Paracoccidioides*, rounded, thick-walled yeast cells can be observed with diameters of 15–30 μm and up to 60 μm in some cases. These may have multiple buds and resemble “ship’s wheel”, “pilot wheel”, or “Mickey Mouse ears”. Histological observations of coccidioidomycosis show tissue with spherules of various sizes (10 to 100 µm) containing numerous endospores [17,24].

For other fungal infections, identification of the etiologic agent based on morphologic characteristics observed in tissue may be problematic. Several studies have assessed the diagnostic accuracy of fungal identification in histopathology specimens compared to culture or other laboratory studies. Overall, concordance between fungal identification by histopathology and either culture or other laboratory studies has ranged between 78% to 95% [32,33,36,37]. In one 10-year retrospective review that included all positive mold and yeast cultures with concurrent surgical specimens (*n* = 47), a correct identification of the etiologic agent was made by histopathology in 37 (78.7%) cases with 10 having discrepant results compared with culture [32]. While several discordant results were due to the misidentification of septate hyaline molds that may mimic each other within tissue (e.g., *Aspergillus* per histopathology, *Scedosporium* or *Fusarium* by culture), others discordant results were between fungi that are generally thought to have distinguishing morphologic characteristics (e.g., *Aspergillus* and the Mucorales). Similar results (78% diagnostic accuracy with cultures) were reported in another small retrospective review (*n* = 58) in which an identification of *Aspergillus* was made based on findings within histology and cytology specimens [36]. Several misdiagnoses by histopathology and cytology were between organisms known to mimic *Aspergillus*, but others included organisms that are thought to have distinguishing morphologies, including the Mucorales, yeasts, and dematiaceous fungi. A more recent study that covered an 18-year period reported a higher concordance (95%) between morphologic diagnosis on histopathology and positive laboratory tests, including antigen tests, serology, and molecular tests in addition to cultures (*n* = 333) [37]. Interestingly, as observed in the previous studies, there were several instances of interpretative errors between *Aspergillus* and the Mucorales. Examples of cases from these studies where identification of the etiologic agent by histopathology differed from culture or other laboratory tests are shown in Table 1. Thus, a reliable identification to the species level, and at times the genus level, based solely on morphologic features on histopathology is not possible.

Unfortunately, an incorrect diagnosis and misidentification of the etiologic agent by histopathology can potentially lead to the use of inappropriate antifungal therapy. For example, voriconazole is considered the drug of choice for the treatment of invasive aspergillosis [38], but the Mucorales, the causative agents of mucormycocis, are intrinsically resistant to this triazole [39,40,41]. Treatment options may also vary between different fungi that are nearly indistinguishable based on solely on histopathologic findings. Certain septate hyaline molds are intrinsically resistant to different antifungals (e.g., *Purpureocillium lilacinum* to amphotericin B, members of the *Rasamsonia argillacea* species complex to voriconazole, isavuconazole, and itraconazole) [42,43,44,45,46,47,48,49]. Moreover, treatment guidelines recommend against the use of specific antifungals when infections are caused by certain species, but not others within the same genera (e.g., amphotericin B is not recommended for *A. terreus* infections but may be used against those caused by *A. fumigatus*) [38,50]. Thus, histopathology findings and the results of laboratory tests are enhanced when used in combination for the diagnosis of invasive fungal infections and the identification of the etiologic agent. Unfortunately, in many instances histopathology findings may be the only results available. In one retrospective study, of the 3164 cases with a morphologic diagnosis by histopathology, only 519 (16%) had concomitant samples submitted for laboratory studies [37].

## 5. Conclusions

Histopathology remains an important early component for the diagnosis of invasive fungal disease. The identifications of the etiologic agents are based on morphologic characteristics, and these findings can offer provisional insights into the likely causes of infections, which can help guide initial therapy while laboratory results are pending. While histopathology can be useful in the differentiation of certain causes of fungal disease (e.g., dematiaceous fungi vs. certain endemic species vs. hyaline molds), care should be taken in the interpretation of these observations. Studies have reported discordant results between histopathology and laboratory results due to overlapping morphologic features. Thus, histopathology can provide a differential of potential fungal pathogens, but identification to the species or even genus level is rarely possible based on these findings alone. Other laboratorial methodologies, including molecular methods which are increasingly being used with tissue specimens, should be used in parallel in order to achieve a rapid and accurate diagnosis of invasive fungal diseases.

## Figures and Tables

**Table 1 jof-08-00505-t001:** Examples of cases from studies where identification of the etiologic agent by histopathology differed from culture or other laboratory tests.

Available Clinical Information [Reference]	Histopathology Diagnosis/ Fungal Identification	Culture Results/ Fungal Identification
Immunocompromised heart transplant recipient (previous antifungal therapy) [32]	Lung biopsy—scant, irregular hyphal forms and rare branching; *Aspergillus* sp.	*Rhizopus* sp.
Immunocompromised patient with relapsed leukemia [32]	Ear canal biopsy—pauciseptate and ribbon-like hyphal elements; suggestive of mucormycosis	*Aspergillus niger*
Immunocompetent child [32]	Lung biopsy with granulomatous inflammation and extracellular ovoid yeast with budding; resembling *Candida* sp.	*Histoplasma capsulatum*
Immunocompetent lung transplant recipient [37]	Endobronchial lung biopsy—fungal organisms consistent with *Candida* sp.	*Aspergillus fumigatus*
Immunocompetent female [37]	Sinus contents—consistent with mucormycosis	*Aspergillus flavus*
Immunocompetent bone marrow transplant recipient [37]	Transbronchial lung biopsy—consistent with *Aspergillus*	*Rhizopus* sp.

## Data Availability

Not applicable.

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
