# Peer review of "Diagnosis from Tissue: Histology and Identification"

_jof, 2022, doi:10.3390/jof8050505_

Round 1

Reviewer 1 Report

The paper is a short review about the role of the histopathology in the ID of fungal diseases.

From my point of view, authors should stand out more clearly at Section 4 that the histopathology does not work to identify fungal pathogens at species level.

Conclusions section should indicate that histopathology techniques can differentiate some cases (some black fungi, some endemic mycosis), but never should be used as reference technique for classification. It should be also noted that molecular methods are the gold standard on tissues showing fungal structures for etiological classification, since culture have a very low sensitivity.

Author Response

We thank the reviewers for their constructive critique and comments, which we believe have improved the quality of this manuscript. Our responses are in attachment.

Reviewer 2 Report

The work of Sabino & Wiederhold is a brief report regarding the utility of histopathology for diagnosis of invasive fungal infections/accuracy for ID of the etiologic agent. In general, the literature review and analysis was properly addressed. Although the manuscript is well written, it lacks originality and relevance.

Author Response

(The authors gave the same response as above.)

Reviewer 3 Report

This is a rather well-written short review. The text has numerous grammatical errors, which I have tried to point out, but I enjoin the authors to go through their manuscript again and check for clarity. However, the main issue is that there seems to be a disconnect between the text in the abstract and the main body text. Please see the attached, annotated PDF. Again, otherwise, this is a fine review.

Author Response

(The authors gave the same response as above.)
